# A Double Negative Feedback Loop between mTORC1 and AMPK Kinases Guarantees Precise Autophagy Induction upon Cellular Stress

**DOI:** 10.3390/ijms20225543

**Published:** 2019-11-07

**Authors:** Marianna Holczer, Bence Hajdú, Tamás Lőrincz, András Szarka, Gábor Bánhegyi, Orsolya Kapuy

**Affiliations:** 1Department of Medical Chemistry, Molecular Biology and Pathobiochemistry, Semmelweis University, 1085 Budapest, Hungary; holczer.marianna@med.semmelweis-univ.hu (M.H.); bbence.hajdu@gmail.com (B.H.); banhegyi.gabor@med.semmelweis-univ.hu (G.B.); 2Laboratory of Biochemistry and Molecular Biology, Department of Applied Biotechnology and Food Science, Budapest University of Technology and Economics, 1111 Budapest, Hungary; tlorincz@mail.bme.hu (T.L.); szarka@mail.bme.hu (A.S.); 3Pathobiochemistry Research Group of the Hungarian Academy of Sciences and Semmelweis University, 1085 Budapest, Hungary

**Keywords:** mTOR, AMPK, autophagy, systems biology, double negative feedback

## Abstract

Cellular homeostasis is controlled by an evolutionary conserved cellular digestive process called autophagy. This mechanism is tightly regulated by the two sensor elements called mTORC1 and AMPK. mTORC1 is one of the master regulators of proteostasis, while AMPK maintains cellular energy homeostasis. AMPK is able to promote autophagy by phosphorylating ULK1, the key inducer of autophagosome formation, while mTORC1 downregulates the self-eating process via ULK1 under nutrient rich conditions. We claim that the feedback loops of the AMPK–mTORC1–ULK1 regulatory triangle guarantee the appropriate response mechanism when nutrient and/or energy supply changes. In our opinion, there is an essential double negative feedback loop between mTORC1 and AMPK. Namely, not only does AMPK downregulate mTORC1, but mTORC1 also inhibits AMPK and this inhibition is required to keep AMPK inactive at physiological conditions. The aim of the present study was to explore the dynamical characteristic of AMPK regulation upon various cellular stress events. We approached our scientific analysis from a systems biology perspective by incorporating both theoretical and molecular biological techniques. In this study, we confirmed that AMPK is essential to promote autophagy, but is not sufficient to maintain it. AMPK activation is followed by ULK1 induction, where protein has a key role in keeping autophagy active. ULK1-controlled autophagy is always preceded by AMPK activation. With both ULK1 depletion and mTORC1 hyper-activation (i.e., TSC1/2 downregulation), we demonstrate that a double negative feedback loop between AMPK and mTORC1 is crucial for the proper dynamic features of the control network. Our computer simulations have further proved the dynamical characteristic of AMPK–mTORC1–ULK1 controlled cellular nutrient sensing.

## 1. Introduction

The maintenance of cellular homeostasis against external and internal stimuli (such as starvation, inflammatory mediators) is controlled by complex protein–protein regulatory pathways. The energy and food supply of a cellular system is directly measured by the AMP/ATP ratio [1,2]. When the ATP level is high, the conditions are optimal for cellular growth, while increasing the level of AMP/ATP quickly blocks the anabolic processes of the cell [2,3]. This mechanism is tightly regulated by the two sensor elements of nutrient conditions and energy supply: one of the mTOR (mammalian target of rapamycin) complexes and AMPK (AMP activated protein kinase) kinase [4,5,6].

mTOR is a serine/threonine protein kinase and is the main component of mTOR pathway-regulated cellular proteostasis by integrating different signals such as growth factors, amino acids, glucose, and energy status to control growth and metabolism [7,8]. For proper function, mTOR kinase has to form a complex, mTORC1, with other regulatory subunits such as regulatory-associated protein of mTOR (Raptor), mammalian lethal with SEC13 protein 8 (MLST8), PRAS40, and DEPTOR [9]. Under physiological conditions, mTORC1 becomes active by a phosphatidylinositol 3 kinase (PI3K)–Akt kinase-dependent manner, while the TSC1-TSC2 (hamartin-tuberin) complex is a critical upstream negative regulator of mTORC1 [8,10]. Inhibition of mTORC1 results in a drastic block of protein synthesis by de-phosphorylating the crucial targets of mTOR such as ribosomal protein S6 kinase (p70S6K) and translation initiation factor 4E binding protein-1 (4E-BP1) [8,10].

Besides mTORC1, AMPK tightly controls ATP-consuming processes such as glycogen or protein syntheses, fatty acids and cholesterol syntheses, and upregulates processes that increase ATP (i.e., glycolysis, β-oxidation) [11,12] by sensing energy levels to maintain cellular homeostasis [13]. AMPK is a heterotrimeric protein complex and it is fairly sensitive to the change in the cellular AMP/ATP ratio. When the AMP level is high in the cell, the free AMP directly binds to AMPK and turns it on [2]. AMPK becomes active when it is phosphorylated at the Thr-172 residue [14].

Both AMPK and mTORC1 control cellular homeostasis via an evolutionary conserved digestive process called autophagy [7,15,16]. During autophagy, cellular components become sequestered into a double membrane vesicle, called an autophagosome, whose contents are then delivered to and degraded by the lysosome [17,18]. Cells always have some residual autophagic activity even under physiological conditions; however, the process becomes more efficient during various stress events (i.e., immune signals, starvation, and growth factor deprivation) to promote the degradation of several cellular components [19,20]. Although autophagy seems to be a cyto-protective process by recycling the harmful and/or unnecessary components of the cell, an excessive level of autophagy might induce cell death [21,22]. One of the key inducers of the complex required for autophagosome formation is Unc-51 like autophagy activating kinase (ULK1/2) [23]. Knock-down of ULK1 is able to inhibit autophagy activation in several cell lines [13,23,24]. After starvation, ULK1/2, together with Fip200 and Atg13, forms a complex that is able to become active through auto-phosphorylation [13]. This process in turn leads to translocation of the entire complex to the pre-autophagosomal membrane and to autophagy induction [13].

AMPK and mTORC1 are known to oppositely control the autophagy inducing complex ULK1/2-Fip200-Atg13 [13]. Under nutrient rich conditions, mTORC1 directly downregulates the self-eating process by phosphorylating various Ser-residues on ULK1 (such as Ser757) [13,16]. In contrast, AMPK has both direct and indirect positive effects on autophagy induction. On one hand, AMPK is able to promote the self-eating mechanism by phosphorylating different phosphorylation sites on ULK1 (e.g., Ser555) in response to nutrient limitation [13,25]. On the other hand, AMPK inhibits the mTOR complex either via the TSC1/2 pathway or by direct phosphorylation of Raptor [13,26,27,28].

In addition, ULK1 is able to inhibit the mTOR complex throughout its Raptor subunit, generating a so called double negative feedback loop (i.e., ULK1 ┤ mTORC1 ┤ ULK1) [13,29,30]. Recently, it has also been shown that ULK1 antagonizes autophagy via downregulating AMPK activity [13,31]. This regulatory connection between AMPK and ULK1 results in a negative feedback loop in the control network. Through ULK1-dependent negative regulation of its up-stream regulators, the control network helps to fine-tune the cellular autophagic response with respond to cellular stress events [13,32].

To understand the dynamic characteristics of the AMPK–mTORC1–ULK1 regulatory triangle, a computational analysis was performed by Szymanska et al. [33]. Their mathematical model was formulated to study the systems-level consequences of interactions among AMPK, mTORC1, and ULK1 by mainly focusing on autophagy induction upon various stress events [33]. By changing the intracellular level of either AMPK (simulating starvation) or mTORC1 (simulating rapamycin treatment) kinases, the regulatory network generated a graded response to stress inputs achieved by a bistable switch between autophagy and protein synthesis [33].

Although Szymanska et al. have already presented a testable mechanistic model of the AMPK–mTORC1–ULK1 regulatory network, they did not take into account many cellular stress events already proven by experimental methods. Here, we directly show that some mutant phenotypes cannot be explained by this simple model (such as ULK1 depletion by using siULK1, or mTORC1 hyper-activation by using siTSC1/2). We claim that mTORC1 is able to downregulate AMPK, generating an extra double negative feedback loop in the control network (i.e., AMPK ┤ mTORC1 ┤ AMPK). By using systems biology techniques, we confirmed that the feedback loops of the AMPK–mTORC1–ULK1 regulatory triangle guarantee the appropriate response mechanism of the cell upon a broad spectrum of external and internal cellular stress effects.

## 2. Results

### 2.1. Downregulation of ULK1 Blocks AMPK Induction

Recently, Szymanska et al. built a mathematical model based on the mTOR–AMPK–ULK1 (from this point mTOR refers to mTORC1) regulatory triangle to explain cellular autophagy induction upon rapamycin treatment or glucose starvation [33]. However, other well-documented mutant behaviors have not been studied with this model yet. To further investigate the systems-level properties of the control network, we created an ordinary differential equation-based mathematical model using their network as a template (see the colored boxes and black lines of the wiring diagram in Figure 1). A double negative feedback has already been described between AMPK and ULK1 (see regulatory connections “a” and “b” in Figure 1); furthermore, mTOR–ULK1 mutual antagonism is also well-known (see regulatory connections “c” and “d” ion Figure 1). AMPK directly inhibits mTOR (see regulatory connection “e” in Figure 1), which blocks autophagy (aka ATG) (see regulatory connection “f” in Figure 1); meanwhile, ULK1 directly promotes ATG activation (see regulatory connection “g” in Figure 1). We assume that when the ATG level is high, that the cellular self-cannibalism is active.

By first running computer simulations, we reproduced glucose starvation and rapamycin treatment (the details of the mathematical code can be found in the Appendix A and also in Appendix A). Next, we tested whether this simple model was also able to describe many other well-known mutant behaviors (these results are summarized in Appendix A). However, we observed that neither modeling cellular ULK1 depletion (ULK1T set to almost zero), nor generating mTOR hyper-activation (2-fold increase of kamtor) could mimic the relevant biological experiment (Appendix A and Appendix A). Namely, both ULK1 depletion or mTOR hyper-activation resulted in a significant increase in the AMPK-P level in the model already at physiological conditions, but no AMPK phosphorylation was actually detected when siULK1 or siTSC1/siTSC2 (aka mTOR hyper-activation) was expressed in human cells [27,34,35,36]. Our preliminary data assumed that one or more important feedback(s) was still missing from the mTOR–AMPK–ULK1 regulatory triangle.

To reveal the missing regulatory connection(s) of this simple control network, we first studied the ULK1 silencing experimentally (Figure 2). ULK1 was directly silenced with siRNA in human embryonic kidney cells (HEK293T), and the efficiency of siULK1 was tested both on the mRNA (data not shown) and protein (Figure 2) levels. To further confirm the effect of ULK1 silencing on autophagy induction, the expression of ULK1 siRNA was also followed by rapamycin (i.e., mTOR inhibition) treatment (100 nM, 2h) or starvation (carbohydrate-free medium, 6 h and 24 h) (i.e., AMPK hyper-activation). At the end of the treatments, the key indicators of autophagy (LC3 and p62) were detected by immunoblotting and the efficiency of the self-eating process was also checked by fluorescence microscope (Figure 2A–C).

After the addition of rapamycin or inducing starvation, a high LC3II/GAPDH ratio and significantly decreased p62 level were observed, suggesting that autophagy was activated and worked properly. When these treatments were preceded by ULK1 silencing, neither the LC3II/GAPDH ratio nor p62 level changed significantly, suggesting that functional autophagy was not detected (Figure 2A,B). HEK293T cells were fixed and immunolabeled for endogenous LC3II/LC3I detection, where LC3 (referring to autophagy) could be observed in discrete foci (Figure 2C). In the control cells treated with rapamycin, a more than 5-fold increase of autophagosomes was detected. In contrast, the relative amount of autophagosomes remained significantly low in both the absence of ULK1 and siULK1 combined with rapamycin treatment. These results fit our western blot analysis and further suggest that ULK1 is essential for autophagy induction.

To explore the activation profile of both AMPK and mTOR in the absence of ULK1, AMPK phosphorylation and the key marker of active mTOR (such as p70S6K-P) was also followed by immunoblotting when ULK1 siRNA was expressed in the cells (Figure 2D). Interestingly, AMPK phosphorylation did not show a significant reduction, while mTOR activity significantly increased when the treatments were carried out in human HEK293T cells expressing siULK1 (see the amount of p70S6K-P in Figure 2D), suggesting that although ULK1-dependent inhibition on AMPK was missing, something else could keep it in its inactive de-phosphorylated state.

Since autophagy was not detected when ULK1 silencing was combined with mTOR inactivation/AMPK hyper-activation, our results further confirm that ULK1 is essential for proper autophagy activation. We also claim that the AMPK-P level cannot increase in the absence of ULK1.

### 2.2. Hyper-Activation of mTOR Blocks AMPK Induction

To explore the role of mTOR kinase in autophagy regulation, mTOR was directly hyper-activated by depleting its stoichiometric inhibitors, called TSC1 and TSC2, respectively (Figure 3). First, the mixture of siTSC1 and siTSC2 was expressed in HEK293T cells. Then, combined treatment was also carried out, when silencing of RNAs was followed by starvation (carbohydrate-free medium, 6 h and 24 h) or rapamycin addition (100 nM, 2 h). The efficiency of siTSC1 and siTSC2 was tested on both the mRNA (data not shown) and protein (Figure 3A,B) levels. At the end of the treatments, the key markers of autophagy (i.e., p62, LC3II/GAPDH) were detected by immunoblotting and the efficiency of the self-eating process was also checked by fluorescence microscope (Figure 3A–C).

Silencing of mTOR inhibitors resulted in a successful hyper-activation of mTOR (see the enhanced phosphorylation of p70S6K in Figure 3A,B). The moderate ratio of LC3II/GAPDH and the high level of p62 suggested that autophagy could not be active (Figure 3A,B). This observation coincided with the assumption that mTOR has a drastic negative effect on autophagy induction. Autophagy induction was also analyzed by using immunofluorescence microscopy with and without TSC1/2 silencing (Figure 3C). Rapamycin-treatment showed an almost 5-fold increase of autophagosomes, while in the absence of TSC1/2, no significant amount of discrete foci was detected. These results suppose that hyper-active mTOR blocks autophagy induction. Interestingly, siTSC1/2 combined with rapamycin treatment resulted in a modest increase of autophagosomes (Figure 3C). These data fit to the immunoblot analysis, where significant increase was observed in LC3II/GAPDH ratio upon treatment (Figure 3A,B), supposing that artificial downregulation of mTOR by rapamycin addition might win against silencing TSC1/2.

In the next step, we also tested the possible influence of mTOR kinase on AMPK activity by following the phosphorylation status of AMPK in cells expressing siTSC1 and siTSC2 (Figure 3D). Although ULK1-controlled inhibition on AMPK was completely missing due to mTOR-dependent downregulation of ULK1 (see regulatory connections “c” and “d” in Figure 1), AMPK remained de-phosphorylated, supposing that AMPK was also inhibited (Figure 3D). In addition, the AMPK-dependent phosphorylation of ULK1 at Ser555 was also not observed (Figure 3D), further supposing that ULK1 was not active.

Taken together, these results confirm that mTOR upregulation keeps autophagy inactive. Although ULK1 activity was not observed due to the hyper-activation of mTOR in the presence of siTSC1 and siTSC2, AMPK remains completely inactive. Since only mTOR is active in this mTOR–AMPK–ULK1 regulatory triangle, we assume that mTOR has some direct or indirect negative effect on AMPK in the absence of ULK1.

### 2.3. The Proper Dynamical Characteristic of ULK1 Depletion Could Be Managed by a Double Negative Feedback Loop between mTOR and AMPK

Our results have shown that phosphorylation-dependent activation of AMPK was not observed when either siULK1 or siTSC1/siTSC2 were expressed in the cells. Since ULK1-dependent inhibition was not present, but mTOR remained active in both treatments (see Figure 2 and Figure 3), we supposed that mTOR might act negatively on AMPK induction. To investigate the importance of this mTOR-dependent inhibition of AMPK on the dynamic properties of our control network, we extended our model with this negative connection (i.e., mTOR ┤ AMPK, see the orange colored line of the wiring diagram in Figure 1). Thereby, an extra double negative feedback loop was generated between mTOR and AMPK (AMPK ┤ mTOR ┤ AMPK).

Mimicking starvation (i.e., induction of AMPK phosphorylation) through computer simulations, AMPK quickly became active and phosphorylated ULK1 (Appendix A). The active AMPK and ULK1 together induced autophagy, however, AMPK-P was later downregulated due to ULK1-dependent de-phosphorylation (Appendix A). In the absence of ULK1 (i.e., silencing of ULK1 by siRNA was mimicked by setting ULK1T = 0.001), the active mTOR level remained high, and due to the mTOR-dependent AMPK inhibition, it was now able to keep AMPK in its inactive dephosphorylated form in this model (Appendix A).

Our computer simulations also nicely fit with our experimental data when mTOR hyper-activation was generated by increasing mTOR activity (kamtor = 0.05). Although hyper-activation of mTOR rapidly downregulated ULK1, AMPK did not become active because mTOR also had a negative effect on AMPK (Appendix A). However, in the absence of the mTOR ┤ AMPK connection, AMPK became active when ULK1 was depleted or mTOR was hyper-activated (see Appendix A), further suggesting the importance of mTOR-dependent downregulation of AMPK.

Our theoretical analysis suggests that mTOR can inhibit AMPK. We suppose that the double negative feedback loop generated by mTOR and AMPK in the control network has a key role in keeping AMPK inactive at physiological conditions. We claim that the level of AMPK-P remains low until mTOR is present.

### 2.4. AMPK Activation Always Precedes ULK1 Induction during Autophagic Stress Response

Since both ULK silencing and mTOR hyper-activation (via TSC1/TSC2 silencing) were carried out under normal growth conditions, we cannot rule out that AMPK did not become active due to unchanged energy status of the cell, rather than via the mTOR-dependent control mechanism. Therefore, to further confirm the importance of the mTOR-dependent downregulation of AMPK-P, the AMPK phosphorylation profile was followed in time when mTOR was inhibited. HEK293T cells were treated with 100 nM rapamycin and the time-dependency of the key markers (i.e., autophagy, AMPK, mTOR, and ULK1) were followed both experimentally (via immunoblotting) and theoretically (via computer simulations) (Figure 4).

Upon rapamycin treatment, the mTOR targets (i.e., p70S6K, 4E-BP1) were quickly downregulated. The active phosphorylated form of p70S6K disappeared, while the third phosphorylation band of 4E-BP1-P appeared, suggesting the complete inactivation of mTOR (Figure 4). AMPK-P could be observed when active mTOR disappeared in the cell, further confirming that mTOR might have some negative effects on AMPK-P. Parallel with the inactivation of mTOR, AMPK became hyper-phosphorylated. AMPK-P was able to promote the phosphorylation and activation of ULK1 (see ULK1-Ser555-P in Figure 4) within a certain delay. However, after two-hour long rapamycin treatment, AMPK-P disappeared due to ULK1-dependent downregulation. Autophagy became active when both AMPK-P and active ULK1 were present in the cell (Figure 4). Although the ratio of LC3II/GAPDH increased after 30 min-long rapamycin treatment, p62 significantly decreased only when ULK1 became phosphorylated (Figure 4). These results suggest that a two-hour long rapamycin treatment is required for dramatic activation of autophagy in HEK293T cells. Interestingly, autophagy remained active at the end of the rapamycin treatment, despite the ULK1-dependent downregulation of AMPK-P (Figure 4), supposing that AMPK-P is not essential to maintain autophagy.

Our results demonstrate that AMPK can be active if mTOR is downregulated, and its activity always precedes ULK1 induction. Although autophagy becomes significantly active only when both AMPK-P and ULK1-P are present in the cell, AMPK-P is not essential to keep the cellular self-cannibalism active.

### 2.5. AMPK Is Not Sufficient to Induce Autophagy during mTOR-Downregulation

Our results clearly suggest that AMPK remains de-phosphorylated until mTOR is active, even in the absence of ULK1 upon various cellular stress events. In order to further confirm this connection, the time-dependency of rapamycin treatment was detected in the absence of ULK1. First, ULK1 gene expression was silenced by siRNA, and then rapamycin treatment (100 nM, for 2 h) was carried out. The key markers (i.e., autophagy, AMPK, mTOR, and ULK1) were followed both experimentally (via immunoblotting) and theoretically (via computer simulations) (Figure 5).

Rapamycin drastically decreased the activity of mTOR after 30 min of treatment. AMPK quickly became significantly phosphorylated one and a half hours later, and remained active in the absence of ULK1 when mTOR was inhibited. These results further confirm that mTOR is crucial to keep AMPK in its inactive de-phosphorylated form when ULK1 is not present in the cell. However, no autophagy induction was observed at all (see the low LC3II/GAPDH ratio and the high p62 level in Figure 5B,C).

Our computer simulations showed that the level of AMPK-P became permanently high while mTOR completely disappeared and ULK1 was silenced (via decreasing the total level of ULK1) (Figure 5D). AMPK became hyper-phosphorylated, suggesting that its negative regulators (i.e., mTOR and ULK1) were missing from the control network. Although AMPK seemed to be active and mTOR was inactive, in the absence of ULK1, no autophagy induction was detected.

Taken together, we can conclude that the AMPK-P level increases when both ULK1 and mTOR are missing from the cell. Although AMPK becomes active, alone is not sufficient to induce autophagy when both ULK1 and mTOR are inhibited. The control network definitively requires ULK1 for autophagy induction.

### 2.6. AMPK Is Essential to Induce Autophagy during mTOR-Downregulation

To further explore the role of AMPK in autophagy induction, we checked the time-dependency of rapamycin treatment, where human cells were pre-treated by Compound C (also called dorsomorphin). Compound C is a widely used cell permeable pyrrazolopyrimidine compound that inhibits a number of kinases including AMPK in a dose-dependent manner [37].

HEK293T cells were pre-treated with 2 µM Compound C for 0.5 h followed by the addition of 100 nM rapamycin, while AMPK, mTOR, ULK1, and autophagy markers were detected by immunoblotting (Figure 6A,B). Compound C completely blocked AMPK activation during the whole treatment, therefore no AMPK phosphorylation was observed. Due to the addition of rapamycin, mTOR was also inhibited (see the intensive dephosphorylation of p70S6K and the appearance of the third phosphorylation band of 4E-BP1-P in Figure 6A,B). In the absence of AMPK-P, no ULK1-Ser555-P phosphorylation was detected, suggesting that ULK1 also remained inactive. The low LC3II/GAPDH ratio and the high p62 level suggest that autophagy is inactive during the treatment.

When the combined addition of both Compound C (via decreasing the rate constant of AMPK activation) and mTOR inhibitor (via decreasing the total level of mTOR) was theoretically tested, ULK1 remained inactive (Figure 6C). Although mTOR-dependent inhibition on ULK1 was missing, ULK1 could not be active because its activation term requires AMPK. Since neither active AMPK nor active ULK1 was detected, autophagy-induced cellular survival could not become active either.

Both experimental and theoretical results indicate that AMPK activation is essential to induce autophagy by promoting ULK1 phosphorylation with respect to various cellular stress events, even in the absence of mTOR.

## 3. Discussion

The maintenance of cellular homeostasis is mainly dependent on the ability of cells to take precise action with respect to various stimuli (e.g., nutrient deprivation). An evolutionary conserved digestive process, called autophagy, has a key role in recycling the damaged or unnecessary cellular components of the cell. Autophagy is tightly controlled by two sensors of nutrient conditions: mTOR and AMPK kinases. mTOR is the master regulator of proteostasis by integrating different external and internal signals, while AMPK senses cellular energy status [4,5,6]. Both kinases control the cellular self-digesting process via ULK1 kinase, one of the inducers of autophagy activator complex [7,15,16]. Many scientific results have already revealed the important feedback loops of the mTOR–AMPK–ULK1 regulatory triangle [13]. In addition, a mathematical model was also built to theoretically describe autophagy induction with respect to rapamycin treatment (mTOR inhibition) or starvation (AMPK activation) [33]. However, we claim that an essential feedback loop is still missing, therefore we re-wired the mTOR–AMPK–ULK1-controlled network (Figure 1) and also tested its dynamic characteristics in response to various stress events incorporating both theoretical and molecular biological techniques.

Our systems biological approach clearly confirmed that AMPK cannot be active when ULK1 is silenced by siRNA (Figure 2 and Appendix A) or mTOR is hyper-activated (via TSC1/TSC2 silencing) (Figure 3 and Appendix A), although the main inhibitor of AMPK (i.e., ULK1) is completely diminished from the cells. In both mutant phenotypes, only mTOR was active, suggesting that mTOR has a direct or indirect negative effect on AMPK. Here, we also revealed that rapamycin-dependent downregulation of mTOR resulted in AMPK phosphorylation (Figure 4), further supposing that mTOR might have a negative effect on AMPK activity directly or indirectly. Since AMPK-dependent inhibition of mTOR is already well-known [13,27], an extra double negative feedback loop might be generated between mTOR and AMPK (see AMPK ┤ mTOR ┤ AMPK in Figure 1). This so called bistable toggle switch is a common feature of biochemical regulatory networks describing an irreversible cellular transition between two well-separated stable states in the control network [38]. Namely, at physiological conditions, mTOR is active and AMPK is inactive, while AMPK quickly becomes activated in the absence of nutrients or energy. In addition, the induction of AMPK results in mTOR inhibition and also promotes autophagy-dependent cellular survival.

Since mTOR, the catalytic element of mTORC1, is a Ser/Thr protein kinase, we suppose that mTOR downregulates AMPK via inhibitory phosphorylation. Therefore, we identified potential Ser and Thr phosphorylation sites on AMPK with NetPhos 3.1 (invented by the Department of Bio and Health Informatics, Technical University of Denmark, Lyngby, Denmark), which is a freely available software, to predict mTOR-dependent serine and threonine phosphorylation sites on AMPK [39]. We found two Ser (Ser-232, Ser-496) and two Thr residues (Thr-356, Thr-488) on AMPK; these phosphorylation motifs represented high similarities to the phosphorylation site preferred by mTOR kinase (for details see Appendix A). This analysis further suggests that mTOR might be able to phosphorylate these Ser and Thr residues on AMPK, supposing a regulatory connection between them. However, this connection must later be proven experimentally.

Although AMPK regulation is already well known, as is its effect on autophagy induction, the precise dynamic characteristics of AMPK-P regulation have not been studied yet. Therefore, here we show, with both experimental and theoretical analysis, the exact role of AMPK in enhancing cellular self-cannibalism. Our data clearly verified that AMPK-P is essential (Figure 6), but is alone not sufficient (Figure 5) to induce autophagy with respect to cellular stress. When rapamycin treatment (i.e., mTOR downregulation) was combined with AMPK-P inhibition (by using Compound C), cells could not turn on autophagy (Figure 6), and ULK1 silencing by siRNA completely blocked autophagy induction even when AMPK-P was present in the absence of mTOR (Figure 5). Since a transient (but not significant) autophagy induction was observed when the addition of Compound C was combined with rapamycin treatment, but ULK1 remained inactive, we further confirmed that AMPK is crucial to promote autophagy induction via ULK1 phosphorylation. To clarify the essential role of AMPK in ULK1 induction, the combined treatment of Compound C (AMPK inhibition) and rapamycin (mTOR inhibition) should be extended by artificial activation of ULK1. If autophagy is active under these experimental conditions, we will be able to prove whether the main role of AMPK-P is to switch on ULK1 upon cellular stress.

Time-dependent mTOR inhibition (via rapamycin treatment) showed that AMPK phosphorylation is observed after mTOR downregulation and always precedes ULK1 induction, while ULK1 is also required to switch on autophagy (Figure 4). Namely, when both AMPK-P and active ULK1 are present in the cell, the self-cannibalism becomes active. Interestingly, the active ULK1 quickly downregulates AMPK in the case of rapamycin treatment (see the quick de-phosphorylation of AMPK after 90 min long rapamycin treatment in Figure 4), while autophagy remains active.

For the proper dynamic characteristics of the control network, this ULK1-dependent downregulation of AMPK assumes a time-delayed negative feedback loop between AMPK and ULK1, otherwise the control system would generate a homeostatic response with respect to cellular stress. Although the negative effect of ULK1 on AMPK activity has already been proven [31], the exact molecular mechanism of this time-delay requires further clarification. We suppose that the de-phosphorylation of AMPK-P is crucial to prevent the hyper-activation of self-cannibalism, which might have a dramatic effect on the cell. Similar results have been already observed in the case of NRF2-dependent downregulation of the AMPK-induced autophagy response during oxidative stress [40,41]. However, these assumptions must be also tested both experimentally and theoretically in the future.

By running computer simulations, our new model of the mTOR–AMPK–ULK1 regulatory triangle was able to describe more than 15 mutant phenotypes (Appendix A), generating the first comprehensive model of the control system. The great advantage of our systems biological approach is that our theoretical analysis does not only describe experimentally already proven cellular treatments, but it is also able to predict the unknown characteristics of the various control elements during treatments. For example, we supposed that siULK1 expression followed by the addition of resveratrol had similar characteristics to the combined treatment of siULK1 + rapamycin (Appendix A). Our computer simulations also suggest that if TSC1/TSC2 silencing is combined with rapamycin treatment, AMPK is significantly phosphorylated (Appendix A, Appendix A) and autophagy also has some activity upon this combined treatment (Figure 3). The active AMPK is able to induce ULK1, suggesting that mTOR downregulation is essential to AMPK activation. Interestingly, hyper-activated mTOR kept inactive AMPK, ULK1, and autophagy when TSC1/TSC2 silencing was combined with the addition of resveratrol or nutrient depletion (Appendix A, Appendix A). We suppose that rapamycin has a stronger effect than resveratrol and nutrient depletion to fight against siTSC1/TSC2-dependent overexpression of mTOR, but this assumption needs further clarification.

To understand how the precise molecular balance of mTOR-AMPK regulates cell survival, in particular autophagy, is highly relevant in several cellular stress related diseases such as neurodegenerative diseases (e.g., Parkinson’s disease, Alzheimer’s disease), metabolic diseases, inflammation, and carcinogenesis [42]. Our data suppose that the mTOR–AMPK–ULK1 regulatory triangle, similar to the mTOR–AMPK–NRF2 regulatory module, might be considered as double-edged swords in various diseases, especially in cancer due to their ambiguous role in promoting cell survival in healthy and cancerous cells [43,44]. Our systems biological analysis definitively made some approach toward improving the understanding of the molecular basis of these complex diseases and might also help to promote advanced therapies against these diseases.

## 4. Materials and Methods

### 4.1. Materials

Rapamycin (Sigma-Aldrich, R0395, St. Louis, MO, USA), DMEM—no glucose, no glutamine (Life Technologies, A14430-01, Carlsbad, CA, USA), and Compound C (Sigma-Aldrich, P5499) were purchased. All other chemicals were of reagent grade.

### 4.2. Cell Culture and Maintenance

As a model system, the human embryonic kidney (HEK293T, ATCC, CRL-3216, Manassas, WV, USA) cell line was used and maintained in DMEM (Dulbecco’s Modified Eagle Medium) (Life Technologies, 41965039) medium supplemented with 10% fetal bovine serum (Life Technologies, 10500064) and 1% antibiotics/antimycotics (Life Technologies, 15240062). Culture dishes and cell treatment plates were kept in a humidified incubator at 37 °C in 95% air and 5% CO_2_.

### 4.3. RNA Interference

RNA interference experiments were performed using Lipofectamine RNAi Max (Invitrogen Thermo Fisher Scientific Inc., Waltham, MA, USA) in GIBCO™ Opti-MEM I (GlutaMAX™-I) Reduced-Serum Medium liquid (Invitrogen) and 20 pmol/mL siRNA. The siTSC1 and siTSC2 oligonucleotides were purchased from Ambion (Ambion Thermo Fisher Scientific Inc., Waltham, MA, USA, 138713, 138741) and the siULK oligonucleotides were purchased from Ambion (118259). A total of 200,000 HEK293T cells were incubated at 37 °C in a CO_2_ incubator in an antiobiotic free medium for 16 h, then the RNAi duplex-Lipofectamine™ RNAiMAX (Invitrogen, 13778-075) complexes were added to the cells overnight. Then, fresh medium was added to the cells and the appropriate treatment was carried out.

### 4.4. SDS-PAGE and Western Blot Analysis

Cells were harvested and lysed with 20 mM Tris, 135 mM NaCl, 10% glycerol, 1% NP40, pH 6.8. Protein content of the cell lysates was measured using the Pierce BCA Protein Assay (Thermo Fisher Scientific Inc., 23225, Waltham, MA, USA). During each procedure, equal amounts of protein were used. SDS-PAGE was done by using Hoefer miniVE (Hoefer Inc., Holliston, MA, USA). Proteins were transferred onto a Millipore 0.45 µM PVDF membrane. Immunoblotting was performed using TBS Tween (0.1%), containing 5% non-fat dry milk or 1% bovine serum albumin (Sigma-Aldrich, A9647) for blocking membrane and for antibody solutions. Loading was controlled by developing membranes for GAPDH or dyed with Ponceau S in each experiment. For each experiment, at least three independent measurements were carried out. The following antibodies were applied: antiLC3B (SantaCruz, sc-271625 Santa Cruz, PRK, USA), antiULK1-Ser555-P (Cell Signaling, 5869S, Danvers, MA, USA), antiULK1 (Cell Signaling, 8054S), antip70S6K-P (Cell Signaling, 9234S), antip70S6K (SantaCruz, sc-9202), antiAMPK-Thr172-P (Cell Signaling, 2531S), antiAMPK (Cell Signaling, 2603S), antiTSC1 (Cell Signaling, 4906S), antiTSC2 (Cell Signaling, 4308S) and antiGAPDH (SantaCruz, 6C5), anti4-EBP1-P (Cell Signaling, 9459S), anti4-EBP1 (Cell Signaling, 9644S), and HRP conjugated secondary antibodies (Cell Signaling, 7074S, 7076S). The bands were visualized using a chemiluminescence detection kit (Thermo Fisher Scientific Inc., 32106).

### 4.5. Immunofluorescence

The cells were transferred to a chamber slide (Nunc™ Lab-Tek™ II Chamber Slide™ System, Thermo Fisher Scientific Inc., 154453PK). After silencing and treatment, the cells were washed in phosphate-buffered saline (PBS) and then fixed in 4% paraformaldehyde for 10 min and washed again with ice-cold PBS three times. The fixed cells were permeabilized with 0.1% Triton-X (in PBS) for 10 min and then washed in PBS three times for 5 min. After permeabilization, the cells were blocked with PBS with 0.1% Tween-20 containing and 3% bovine serum albumin for 30 min and incubated overnight with antiLC3A/B (Cell Signaling, 4108S) in 1% bovine serum albumin containing PBS at 4 °C. Cells were washed with cold PBS and incubated with Alexa Fluor 488 conjugated anti-rabbit (Cell Signaling, 4412S) for 1 h. Cells were washed in PBS and treated with DAPI (1:10000) for 15 min and washed again twice for 3 min. The slides were mounted with FluorSave Reagent (Millipore, 345789, Burlington, MA, USA) and observed under a fluorescence microscope (Nikon Eclipse Ts2R, Minato, Tokyo, Japan).

### 4.6. Mathematical Modeling

The regulatory network given in Figure 1 was translated into a set of nonlinear ordinary differential equations (ODEs) and analyzed using the techniques of dynamical system theory [38,45,46]. Dynamical simulations, phase plane, and bifurcation analysis were carried out using the program *XPPAUT,* which is freely available from http://www.math.pitt.edu/~bard/xpp/xpp.html [38,46]. ODE describes the time-rate of the change of level or activity of a component (such as AMPK, ULK1, mTOR, and autophagy inducer (ATG)). The initial conditions and parameter values used for the simulations are given in the Appendix A. All variables are dimensionless; rate constants (*k’s)* have a dimension of min^-1^, while Michaelis Menten constants are dimensionless. The total level of AMPK, ULK1, mTOR, and ATG were assumed to be constant (one unit). We provide the *XPP* code (in the Appendix A) that was used to generate all the figures in the manuscript.

### 4.7. Statistics

For densitometry analysis, western blot data were acquired using ImageJ software (https://imagej.net/). The relative band densities were shown and normalized to an appropriate total protein or GAPDH band used as the reference protein (see Appendix A). For each of the experiments, three independent measurements were carried out. Results were presented as the mean values ± S.D. and were compared using ANOVA with Tukey’s multiple comparison post hoc test. Asterisks indicate statistically significant difference from the appropriate control: ns—nonsignificant; * *p* < 0.05; ** *p* < 0.01.

## Figures and Tables

**Figure 1 ijms-20-05543-f001:**
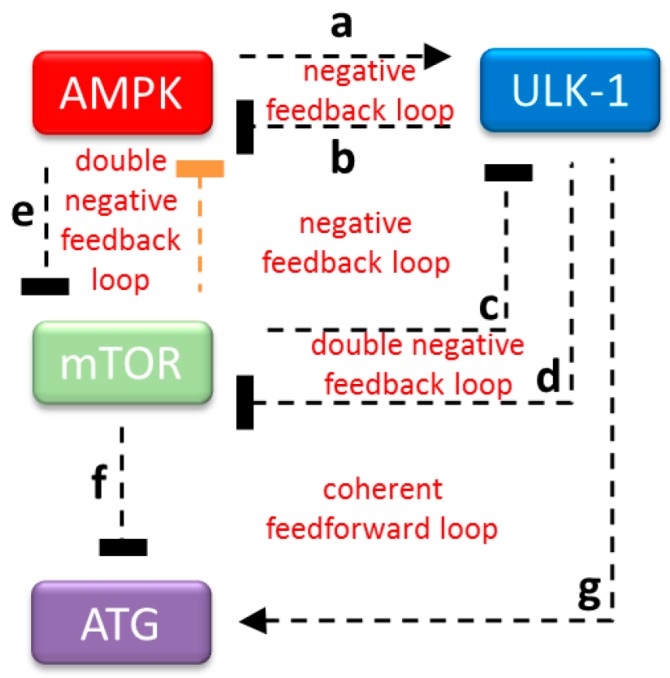
The wiring diagram of the AMPK–mTOR–ULK1 regulatory triangle controlled stress response mechanism. AMPK, mTOR, and ULK1 are denoted by isolated red, green and blue boxes, respectively. ATG (see purple box) refers to the autophagy activator complex. Dashed line shows how the components can influence each other, while blocked end lines denote inhibition. The feedback loop proven here both experimentally and theoretically is orange-colored.

**Figure 2 ijms-20-05543-f002:**
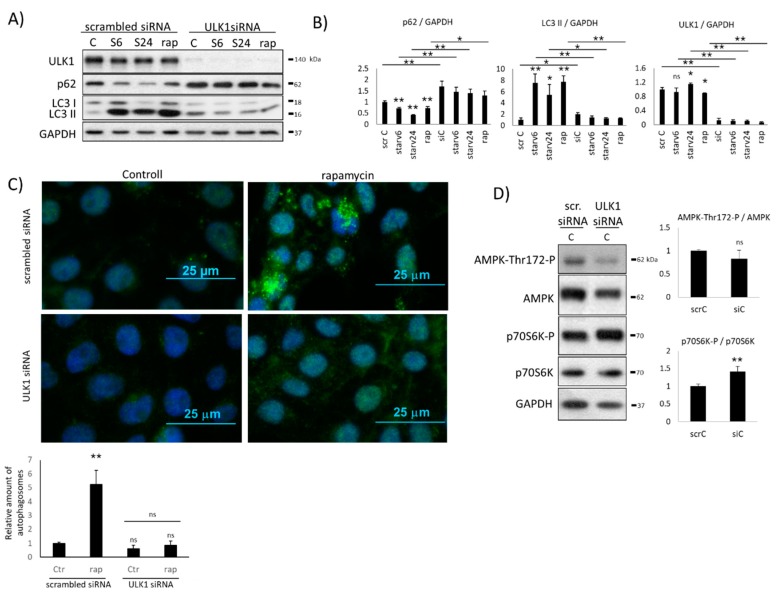
The effect of ULK1 silencing on the AMPK level under various stress events. HEK293T cells were starved (for 6h or 24h) or treated with rapamycin (rap – rapamycin —100 nM, 2 h) without/with silencing of ULK1 by siRNA. (**A**) The markers of autophagy (LC3, p62) and ULK1 were followed by immunoblotting. GAPDH was used as the loading control. (**B**) Densitometry data represent the intensity of p62, ULK1, and LC3 II normalized for GAPDH. (**C**) The connection of ULK1 silencing and autophagy induction was checked by immunofluorescence microscopy. LC3 was stained by green fluorescence dye. Rapamycin (rap–rapamycin—100 nM, 2h) was used as the positive control. Quantification and statistical analysis of the immunofluorescence microscopy data. Error bars represent standard deviation, asterisks indicate statistically significant difference from the control: ns—nonsignificant; * *p* < 0.05; ** *p* < 0.01, (**D**). The markers of AMPK and mTOR (p70S6K-P) were followed in HEK293T cells without/with silencing of ULK1 by siRNA. Densitometry data represent the intensity p70S6K-P normalized for the total level of p70S6K and AMPK-Thr172-P normalized for the total level of AMPK. For each of the experiments, three independent measurements were carried out. Error bars represent standard deviation, asterisks indicate statistically significant difference from the control: ns—nonsignificant; * *p* < 0.05; ** *p* < 0.01.

**Figure 3 ijms-20-05543-f003:**
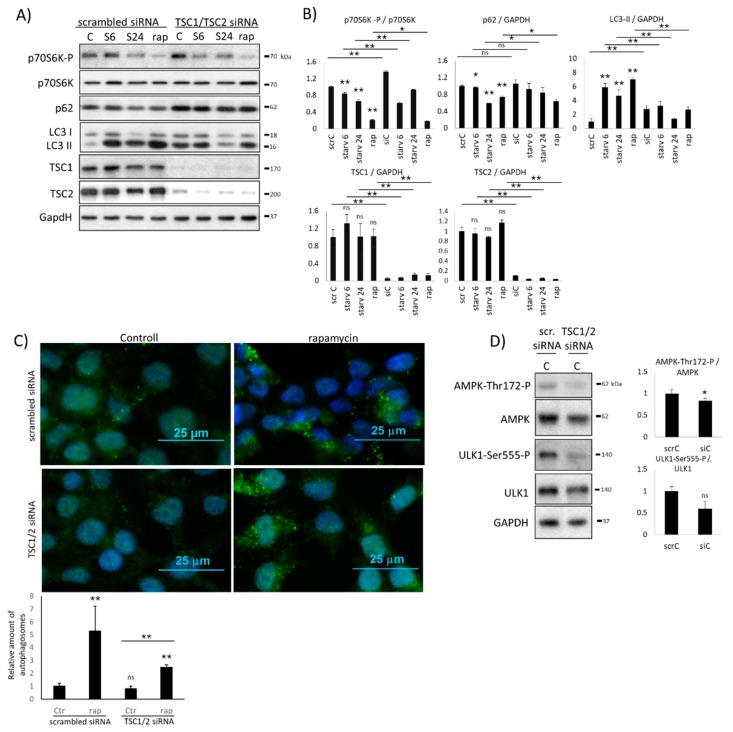
The effect of mTOR hyper-activation (by TSC1/2 silencing) on AMPK level under various stress events. HEK293T cells were starved (for 6h or 24h) or treated with rapamycin (rap – rapamycin —100 nM, 2 h) without/with silencing of TSC1/2 by siRNA. (**A**) The markers of autophagy (LC3, p62), mTOR (p70S6K-P), TSC1, and TSC2 were followed by immunoblotting. GAPDH was used as the loading control. (**B**) Densitometry data represent the intensity of p62 LC3 II, TSC1, and TSC2 normalized for GAPDH, and p70S6K-P normalized for the total level of p70S6K. (**C**) The connection of ULK1 silencing and autophagy induction was checked by immunofluorescence microscopy. LC3 was stained by green fluorescence dye. Rapamycin (rap – rapamycin —100 nM, 2h) was used as the positive control. Quantification and statistical analysis of immunofluorescence microscopy data. Error bars represent standard deviation, asterisks indicate statistically significant difference from the control: ns—nonsignificant; * *p* < 0.05; ** *p* < 0.01, (**D**) The markers of AMPK and ULK1 (ULK1-Ser555-P) were followed in HEK293T cells without/with silencing of TSC1/2 by siRNA. Densitometry data represent the intensity and AMPK-Thr172-P normalized for the total level of AMPK and ULK1-Ser555-P normalized for the total level of ULK1. For each of the experiments, three independent measurements were carried out. Error bars represent standard deviation, asterisks indicate statistically significant difference from the control: ns—nonsignificant; * *p* < 0.05; ** *p* < 0.01.

**Figure 4 ijms-20-05543-f004:**
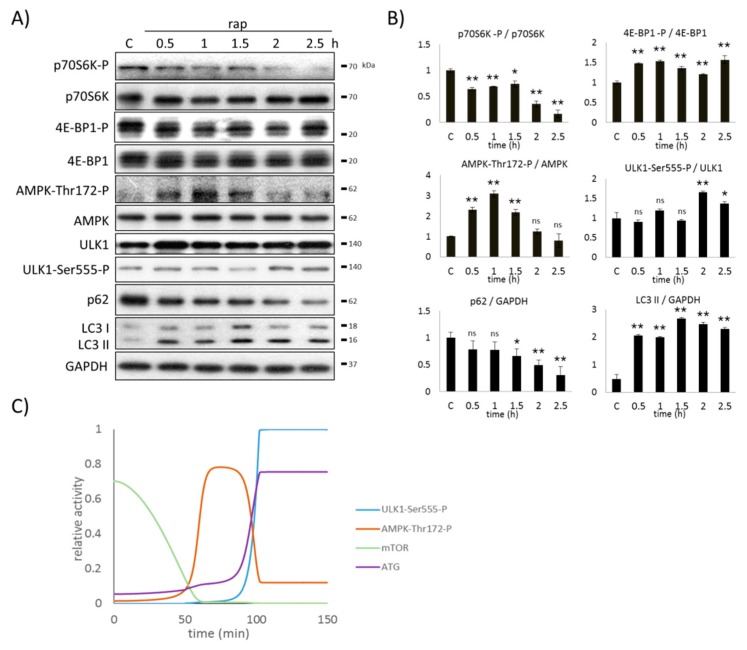
AMPK induction precedes ULK1 activation upon rapamycin treatment. HEK293T cells were denoted in time after 100 nM rapamycin treatment. (**A**) The markers of autophagy (LC3, p62), AMPK, ULK1 (ULK1-Ser555-P), and mTOR (p70S6K-P, 4E-BP1-P) were followed by immunoblotting. GAPDH was used as the loading control. (**B**) Densitometry data represent the intensity of p62 and LC3 II normalized for GAPDH, ULK1-Ser555-P normalized for the total level of ULK1, p70S6K-P normalized for the total level of p70S6K, 4E-BP1-P normalized for the total level of 4E-BP1, and AMPK-Thr172-P normalized for the total level of AMPK. For each of the experiments, three independent measurements were carried out. Error bars represent standard deviation, asterisks indicate statistically significant difference from the control: ns—nonsignificant; * *p* < 0.05; ** *p* < 0.01. (**C**) Computer simulation of rapamycin treatment. The relative activity of AMPK-Thr172-P, mTOR, ULK1-Ser555-P and autophagy (ATG) are plotted in time.

**Figure 5 ijms-20-05543-f005:**
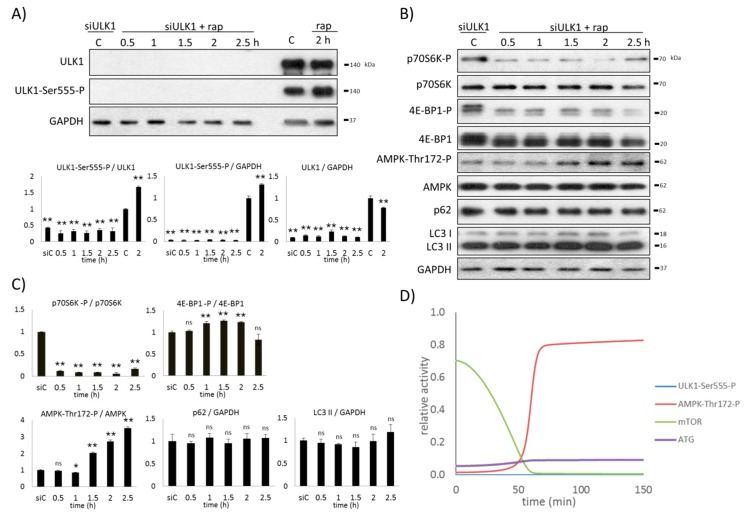
Silencing of ULK1 arrests autophagy induction upon rapamycin treatment. ULK1 was silenced by siRNA in HEK293T cells, followed by 100 nM rapamycin treatment in time. (**A**) Test of the ULK1 silencing by immunoblotting, total level of ULK1 and ULK1-Ser555-P were followed. GAPDH was used as the loading control. Densitometry data represent the intensity of ULK1 and ULK1-Ser555-P normalized for GAPDH and ULK1-Ser555-P normalized for total level of ULK1. (**B**) The markers of autophagy (LC3, p62), AMPK, and mTOR (p70S6K-P, 4E-BP1-P) were followed by immunoblotting. GAPDH was used as the loading control. (**C**) Densitometry data represent the intensity of p62 and LC3 II normalized for GAPDH, p70S6K-P normalized for the total level of p70S6K, 4E-BP1-P normalized for the total level of 4E-BP1, and AMPK-Thr172-P normalized for the total level of AMPK. For each of the experiments, three independent measurements were carried out. Error bars represent standard deviation, asterisks indicate statistically significant difference from the control: ns—nonsignificant; * *p* < 0.05; ** *p* < 0.01. (**D**) Computer simulation of rapamycin treatment. The relative activity of AMPK-Thr172-P, mTOR, ULK1-Ser555-P and autophagy (ATG) are plotted in time.

**Figure 6 ijms-20-05543-f006:**
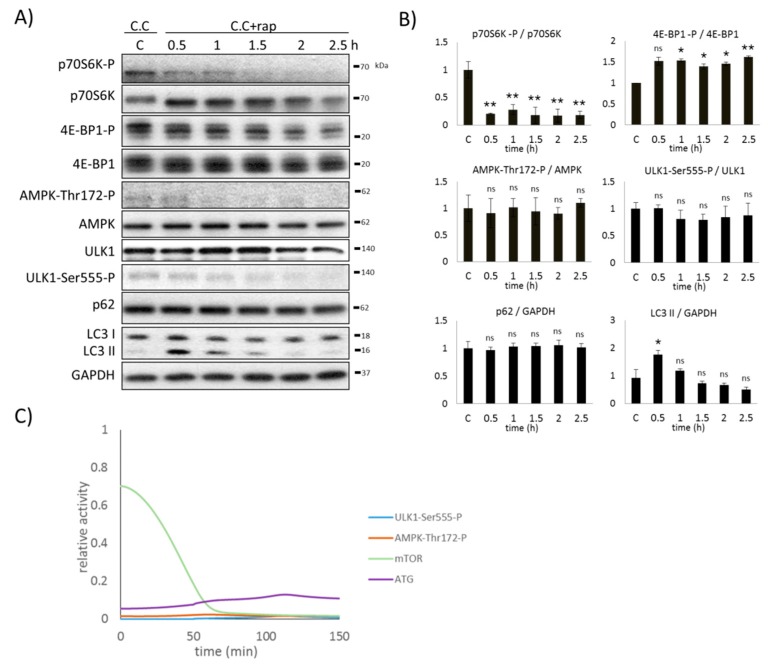
Inhibition of AMPK arrests ULK1 induction upon rapamycin treatment. HEK293T cells were pre-treated with the AMPK inhibitor, Compound C (2 µM, 0.5 h), followed by 100 nM rapamycin treatment in time. (**A**) The markers of autophagy (LC3, p62), AMPK, ULK1 (ULK1-Ser555-P), and mTOR (p70S6K-P, 4E-BP1-P) were followed by immunoblotting. GAPDH was used as the loading control. (**B**) Densitometry data represent the intensity of p62 and LC3 II normalized for GAPDH, ULK1-Ser555-P normalized for the total level of ULK1, p70S6K-P normalized for the total level of p70S6K, 4E-BP1-P normalized for the total level of 4E-BP1, and AMPK-Thr172-P normalized for the total level of AMPK. For each of the experiments, three independent measurements were carried out. Error bars represent standard deviation, asterisks indicate statistically significant difference from the control: ns—nonsignificant; * *p* < 0.05; ** *p* < 0.01. (**C**) Computer simulation of rapamycin treatment. The relative activity of AMPK-Thr172-P, mTOR, ULK1-Ser555-P, and autophagy (ATG) are plotted in time.

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
