# Peer review of "A Double Negative Feedback Loop between mTORC1 and AMPK Kinases Guarantees Precise Autophagy Induction upon Cellular Stress"

_ijms, 2019, doi:10.3390/ijms20225543_

Round 1
Reviewer 1 Report
A computational analysis of a mTORC1, AMPK1 and ULK1 based switch has been previously reported by Szymanska P. and colleagues (PLOS 2015).
In the manuscript titled "A double negative feedback loop between mTOR and AMPK kinases guarantees a precise autophagy induction upon cellular stress" Holzczer M. and colleagues, by means of a combined
approach, forecasting theoretical (computer simulation) and experimental (wet bench research) data, they assessed a double negative feedback loop occurring between mTORC1 and AMPK.
Thereby, the major novelty of the present manuscript lies in providing evidence useful to define such double negative feedback between AMPK and mTORC1 that is required to properly trigger autophagy.
On the whole I appreciated the manuscript. However, prior publication there are some major concerns that have to be sorted-out.
Major concerns
1) English need to be seriously and thoroughly revised especially in the style.
2) Figures need to be arranged differently. Why? There are at least couple of things that need to be improved: a) densitometry data are plotted but when printed-out the labelling is barely readable; 2) IF quantification plots (Fig. 2 & Fig. 3) need to be placed either alongside the IF picture, or below it. Usually, they are just below IF but in tandem with a WB and other quantification data. This kind of arrangement is somewhat ...
confusing.
3) Fig. 3 panel C: According to the plot, rapamycin treatment leads to an impaired autophagy in those cells in which TSC1/2 has been silenced. However the picture does not seem to be on line with these data.
Please clarify this issue.
4) Page 4 par. 2.4 "AMPK activation always precedes ULK1 induction during autophagic stress response". According to your data this is true when cells are treated with rapamycin in which the
oscillatory AMPK kinetic preceds the delayed ULK activation. However, resveratrol does not trigger the same kind of kinetic. In the latter case AMPK and ULK activation seem to have overlapping kinetic.
How is this behaviour explained?
5) the authors mention that the mTORC1 is a crucial for cellular proteostasis. It would be appreciated very much if the authors could check 4E-BP1 phospholrylation status in parallel to that of S6K.
Minor issues
1) Please formatting references according guidelines (e.g. pag. 4 line 137)
2 Fig. or Figure? Please keep it uniform within the manuscript.
3) Within the manuscript there are inaccuracies that need to be edited (e.g. pag.2 lines 44-46: to carry-out its kinase activity the TOR kinase needs to be complexed with different regulators
to form two different kind of complexes namely mTORC1 and mTORC2, where the first one acts as nutrient sensor and not the TOR kinase by itself; pag. 2 lines 53-54 "TSC1-TSC2 (hamatin-tuberin) complex is
a critical downstream negative regulator of mTORC1")
4) I would prefer state in the title mTORC1 and not simply mTORC, since in the manuscript mTORC2 has not been investigated at all.
5) In figures specify which is the AMPK P site
6) Typo mistakes (e.g. Rapamycin vs rapamycin)
Author Response
A computational analysis of a mTORC1, AMPK1 and ULK1 based switch has been previously reported by Szymanska P. and colleagues (PLOS 2015).
In the manuscript titled "A double negative feedback loop between mTOR and AMPK kinases guarantees a precise autophagy induction upon cellular stress" Holzczer M. and colleagues, by means of a combined
approach, forecasting theoretical (computer simulation) and experimental (wet bench research) data, they assessed a double negative feedback loop occurring between mTORC1 and AMPK.
Thereby, the major novelty of the present manuscript lies in providing evidence useful to define such double negative feedback between AMPK and mTORC1 that is required to properly trigger autophagy.
On the whole I appreciated the manuscript. However, prior publication there are some major concerns that have to be sorted-out.
Major concerns
1) English need to be seriously and thoroughly revised especially in the style.
The text has been thoroughly revised, grammatical and syntax errors have been amended.
2) Figures need to be arranged differently. Why? There are at least couple of things that need to be improved: a) densitometry data are plotted but when printed-out the labelling is barely readable; 2) IF quantification plots (Fig. 2 & Fig. 3) need to be placed either alongside the IF picture, or below it. Usually, they are just below IF but in tandem with a WB and other quantification data. This kind of arrangement is somewhat ...
confusing.
Densitometry plots have been corrected (see Figure 2, 3, 4, 5 and 6) and Figure 2 and 3 have been re-arranged. Now the IF quantification plots are placed below the IF pictures. We really hope that this arrangement makes clear and easily understandable our figures.
3) Fig. 3 panel C: According to the plot, rapamycin treatment leads to an impaired autophagy in those cells in which TSC1/2 has been silenced. However the picture does not seem to be on line with these data.
Please clarify this issue.
We agree with the reviewer that the plots, IF data and the computational simulations seem to be a bit contradictory. Our IF data suggest, that the amount of discrete foci of LC3 is increasing when TSC1/2 silencing is combined with rapamycin treatment. However, WB data suppose that this increased amount of LC3 results only in an impaired autophagy, since LC3/GAPDH and p62 levels did not change significantly (see Figure 3B). Our computational simulation is misleading in the supplementary information (Fig. S2D) where a perfect autophagy activation is plotted. Re-calculating our data we realized that TSC1/2 silencing was not properly silenced in our model. Therefore the simulation was repeated with lower level of TSC1/2 (i.e. higher level of mTOR) resulting in an impaired autophagy induction (see the revised Fig. S2D).
According to the new simulation the text, the supplementary information and figures have been revised.
4) Page 4 par. 2.4 "AMPK activation always precedes ULK1 induction during autophagic stress response". According to your data this is true when cells are treated with rapamycin in which the
oscillatory AMPK kinetic preceds the delayed ULK activation. However, resveratrol does not trigger the same kind of kinetic. In the latter case AMPK and ULK activation seem to have overlapping kinetic.
How is this behaviour explained?
We claim that resveratrol treatment is similar to rapamycin addition, but not exactly the same. Rapamycin has a “drastic” inhibitory effect on mTOR pathway, while resveratrol has a milder inhibitory effect on mTOR, but it also enhances AMPK. Therefore the effect of resveratrol is not as severe, i.e. 2.5 hours long rapamycin treatment quickly induced a transient AMPK phosphorylation followed by ULK1 phosphorylation and autophagy induction. However scientific results have already shown that the effect of resveratrol can be realized much slower, therefore the treatment has to be much longer (24 h long). A significant AMPK phosphorylation occurs after 8 h, while ULK1 phosphorylation can be detected not earlier than 16 h. We think that resveratrol treatment requires more than 24 h to show AMPK de-phosphorylation referring to its transient characteristic, while ULK1 remains phosphorylated. We also claim that AMPK does not get fully dephosphorylated due to resveratrol-dependent AMPK activation. However, according to the simulation we suppose that resveratrol treatment shows similar dynamical characteristic to rapamycin treatment.
5) the authors mention that the mTORC1 is a crucial for cellular proteostasis. It would be appreciated very much if the authors could check 4E-BP1 phospholrylation status in parallel to that of S6K.
mTORC1 substrate 4-EBP1-P phosphorylation status has been shown in the figures of revised version (see Figure 4, 5 and 6). Parallel to S6K phosphorylation/de-phosphorylation status, the change of 4-EBP1 phosphorylation status upon various treatments further confirmed our results.
Minor issues
1) Please formatting references according guidelines (e.g. pag. 4 line 137)
The references have been corrected.
2) Fig. or Figure? Please keep it uniform within the manuscript.
„Figure” has been used everywhere in the revised version.
3) Within the manuscript there are inaccuracies that need to be edited (e.g. pag.2 lines 44-46: to carry-out its kinase activity the TOR kinase needs to be complexed with different regulators
to form two different kind of complexes namely mTORC1 and mTORC2, where the first one acts as nutrient sensor and not the TOR kinase by itself; pag. 2 lines 53-54 "TSC1-TSC2 (hamatin-tuberin) complex is
a critical downstream negative regulator of mTORC1")
The text has been thoroughly revised and edited.
4) I would prefer state in the title mTORC1 and not simply mTORC, since in the manuscript mTORC2 has not been investigated at all.
“mTORC1” has been used in the title of the revised version.
5) In figures specify which is the AMPK P site
AMPK-P site has been specified in figures of the revised version.
6) Typo mistakes (e.g. Rapamycin vs rapamycin)
Typo mistakes have been thoroughly revised and fixed.
Reviewer 2 Report
Dear Editor,
The corresponding authors well studies the autophagy under the celluar stress conditions by using various models. The present article is well written of their results and acceptable to publish with minor grammatical changes.
Author Response
Dear Editor,
The corresponding authors well studies the autophagy under the celluar stress conditions by using various models. The present article is well written of their results and acceptable to publish with minor grammatical changes.
We really thank to the reviewer the positive comments. The text has been thoroughly revised, grammatical and syntax errors have been amended.
Round 2
Reviewer 1 Report
The revised form of the manuscript looks better. However, I am still a little bit concerned about the rapamycin vs resveratrol kinetics. I understand that these two compounds display different kinetics (being resveratrol slower than rapamycin) because of their diverse mechanisms of action (resveratrol inhibits mTOR through ATP competition) and that the effects of resveratrol are milder and slower when compared to rapamycin. Since the data refer to a 24 hrs. time-window the conclusions drawn by the authors are too much speculative and weakly supported by the data. In their rebuttal letter, the authors themselves “…simulation we suppose that resveratrol treatment shows similar dynamical characteristics to rapamycin treatment” Thereby, I would prefer either skip this part (fig. S3) or, alternatively, prolong the kinetic up to 48-60 hrs.
Author Response
We thank the comments to the reviewer.
We agree with the reviewer that our dynamical desciption of resveratrol treatment is weakly supported by the data, therefore Figure S3 and the relevant texts are removed from the new version of the manuscript.